# BrainTACO: an explorable multi-scale multi-modal brain transcriptomic and connectivity data resource
Florian Ganglberger [1,2], Dominic Kargl[3,5], Markus Töpfer[1,5], Julien Hernandez-Lallement[2], Nathan Lawless [2], Francesc Fernandez-Albert[2], Wulf Haubensak [3,4] & Katja Bühler [1]✉

Exploring the relationships between genes and brain circuitry can be accelerated by joint analysis of heterogeneous datasets from 3D imaging data, anatomical data, as well as brain networks at varying scales, resolutions, and modalities. Generating an integrated view, beyond the individual resources' original purpose, requires the fusion of these data to a common space, and a visualization that bridges the gap across scales. However, despite ever expanding datasets, few platforms for integration and exploration of this heterogeneous data exist. To this end, we present the BrainTACO (Brain Transcriptomic And Connectivity Data) resource, a selection of heterogeneous, and multi-scale neurobiological data spatially mapped onto a common, hierarchical reference space, combined via a holistic data integration scheme. To access BrainTACO, we extended BrainTrawler, a web-based visual analytics framework for spatial neurobiological data, with comparative visualizations of multiple resources. This enables gene expression dissection of brain networks with, to the best of our knowledge, an unprecedented coverage and allows for the identification of potential genetic drivers of connectivity in both mice and humans that may contribute to the discovery of dysconnectivity phenotypes. Hence, BrainTACO reduces the need for time-consuming manual data aggregation often required for computational analyses in script-based toolboxes, and supports neuroscientists by directly leveraging the data instead of preparing it.

An increasing amount of evidence suggests that human behaviors, and their impairments in psychiatric disorders, are better understood via a multimodal data integration approach than by analyzing individual neurobiological measures[1,2]. Many insights into the brain's functional organization and neuronal mechanisms were sparked by collecting and interpreting spatially organized histology, cellular composition, connectivity, and activity data. For instance, the entry points for modern neuroscientific experimental workflows are brain regions (i.e. part of a specific neuronal circuit thought to be involved in a brain function or behavior) whose gene expressions and functional connectivity patterns are studied to understand the circuit dynamics underlying a behavior. That information can then be used to identify targets in the brain that could be modulated by psychoactive drugs, in cases of psychiatric symptomatology[3]. Thus, integrating both functional connectivity and omics data modalities is instrumental to better understanding the biological underpinnings of behaviors and their deficits[4].

Recent advances in neuroimaging allowed big brain initiatives and consortia to create vast resources[5–7] of data, which could be mined for additional and deeper insights. However, collecting these data from different sources for comparison and exploration leads to several challenges, as they are acquired in different systems, and can vary in resolution, anatomical scale, or sampling density. A mandatory first step is to map the data onto a common reference space, to ensure alignment (for imaging data) and annotation using the same brain region ontology[8]. In an alternative approach, such as mapping the data to the smallest common denominator, e.g. major anatomical brain regions[9], one loses granularity and specificity, rendering the data potentially less representative.

Neuroscience studies that use a combination of omics, imaging, anatomical, and connectivity data often require extensive analytical workflows, including mapping to a common reference space[8], manual data aggregation[9], and statistical analysis. This typically requires the expertise of a bioinformatician to find patterns that might relate to a given behavior[9–16].

---

[1]Biomedical Image Informatics, VRVis Research Center, Vienna, Austria. [2]Global Computational Biology and Digital Sciences, Boehringer Ingelheim Pharma, Biberach an der Riss, Germany. [3]Department of Neuronal Cell Biology, Vienna Medical University, Vienna, Austria. [4]Research Institute of Molecular Pathology (IMP), Vienna Biocenter (VBC), Vienna, Austria. [5]These authors contributed equally: Dominic Kargl, Markus Töpfer. ✉e-mail: buehler@vrvis.at

The term "big" refers to the amount (vast image collections, many datasets) and/or size (high-resolution imaging/network data) of the data, which is too extensive to be analyzed with traditional methods. Here, visual analytics tools bridge this gap by enabling neuroscientists to interactively browse vast data collections, visualize complex relationships, and link different types of data.

Many neuroscientific resources for transcriptomic data provide interactive, web-based visualizations for access and exploration. A comprehensive collection of such websites has been provided by Keil et al.[17]. While providing access to scientists without the need for advanced computational expertise, they are primarily suited for single datasets, i.e., they rarely provide workflows across multiple datasets and modalities. One notable exception is the SIIBRA-Explorer via EBRAINS[18] which combines structural connectivity (fiber tracts)[19] with microarray-based gene expression data[20,21]. Another relevant tool, although not web-based, is BrainExplorer[22,23], which enables the retrieval of structural connectivity from the Allen Mouse Brain Connectivity Atlas[24] in combination with in-situ-hybridization data[25]. Nevertheless, in their current state both, SIIBRA-Explorer and BrainExplorer, are limited to one connectivity and one transcriptomic dataset each, while lacking support for next-generation sequencing data.

In this paper, we present a holistic data integration scheme to map heterogeneous brain data across scales, spatial and anatomical resolutions, as well as sampling and acquisition types (Fig. 1). BrainTACO (Brain Transcriptomic And Connectivity Data) is a resource that includes bulk and single-cell/nucleus RNA sequencing, in-situ hybridization, and microarray-based transcriptomics data, as well as structural and functional connectivity mapped onto common hierarchical reference spaces. To make BrainTACO accessible, we built onto previous work, BrainTrawler[26], a tool for visualizing volumetric, geometry, and connectivity data simultaneously in 3D rendering and 2D slice views, which can iteratively integrate additional heterogeneous datasets from the community and across species. We extended BrainTrawler to integrate, store and query datasets from various resources. Via our spatial indexing-based data structure[27], it enables automatic aggregation and interactive exploration of large-scale, high-resolution spatial connectivity[7,24], and image collections of gene expression data[25] on different scales. We extended our data structure to integrate sample-based region-level datasets (i.e. sampled from brain regions), such as microarray gene expression data or count matrices from RNA sequencing. Here, it is possible to aggregate samples on individual dataset-level by user-defined regions of interest in real-time, so that different datasets can be compared on the same anatomical level, independent of their original resolution and scale.

Via a web interface, BrainTACO can be used to dissect brain connectivity interactively with a wealth of transcriptomic data, similar to Ganglberger et al.'s[26] previous approach for in-situ hybridization data only. To account for the increased number of datasets, as well as the increased complexity of the datasets itself (e.g. samples from multiple cell types, developmental states, etc.), we added additional comparative exploration functions. Here, we facilitated visualization techniques such as heatmaps, small multiples[28], and parallel coordinates to identify gene expression patterns across datasets and categorical information (cell types, phenotypes, developmental stage) interactivity on arbitrary levels of anatomical detail. This enables neuroscientists a view on the data, tailored to their research focus and without the need for programming knowledge.

Our resource closes the gap in current interactive analytical tools by combining gene expression, structural, and functional relationships at the microscopic, mesoscopic, and macroscopic level. This is achieved by the following:

- A hierarchical brain ontology-based integration scheme (i.e. brain parcellation with standardized hierarchical region annotation) to access neurobiological, spatially mapped data across resolution, anatomical scale, or sampling density.
- A collection of publicly available gene expression (in-situ hybridization, microarray, bulk and single-cell/nucleus RNA sequencing) and connectivity (structural and functional resting-state) datasets covering major anatomical brain regions mapped onto common hierarchical reference spaces. The data's original annotation is stored and made transparent (data provenance).
- An intuitive web interface for comparative visualization to access the BrainTACO resource in real-time without programming knowledge.

## Results
### Integrating multi-modal multi-scale resources
To create a resource of brain-wide gene expression and connectivity, we mapped heterogeneous neurobiological spatial datasets to common mouse[29] and human[30] reference spaces. We included a range of single-cell/nucleus RNA sequencing datasets (Fig. 2) covering both species. While the datasets were representative of the whole mouse brain[31–37], the gaps in human data (e.g. Amygdala, Thalamus, Hypothalamus)[9,35,38,39] were filled using bulk RNA sequencing datasets (Fig. 2, GTEx and BrainSpan[40,41]). The included datasets were selected to cover a diversity of meta information, such as morpho-electric cell types (patch sequencing[35]), age information (BrainSpan[41], Battacherjee at al.[34], Lee et al.[39], and the STAB datasets[9,38,42–52]), and different treatment groups (Rossi et al.[33] and Battacherjee at al.[34]). To increase spatial resolution, we added in-situ hybridization data (200-μm voxel-level resolution)[25] and microarray gene expression data for 3702 biopsy sites[20,21], both already mapped to the reference spaces.

Fig. 1 | **Mapping data of different resolution and scales to a common reference space.** (1) Voxel-level data is mapped to the voxel-level reference space by image registration. (2) Region-level data (e.g. RNA-sequencing data) is mapped via a hierarchical brain region ontology (i.e. standardized hierarchical brain parcellation with region annotation) with voxel-level parcellation to the reference space. (3) All data mapped to the reference space can be either retrieved on the resolution of the reference space (data is up- or downsampled via nearest-neighbor interpolation), or on every other region level of the hierarchy.

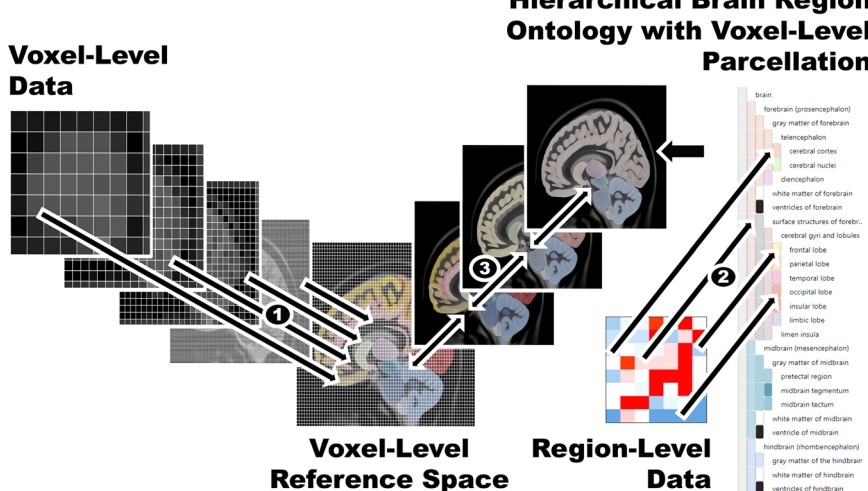

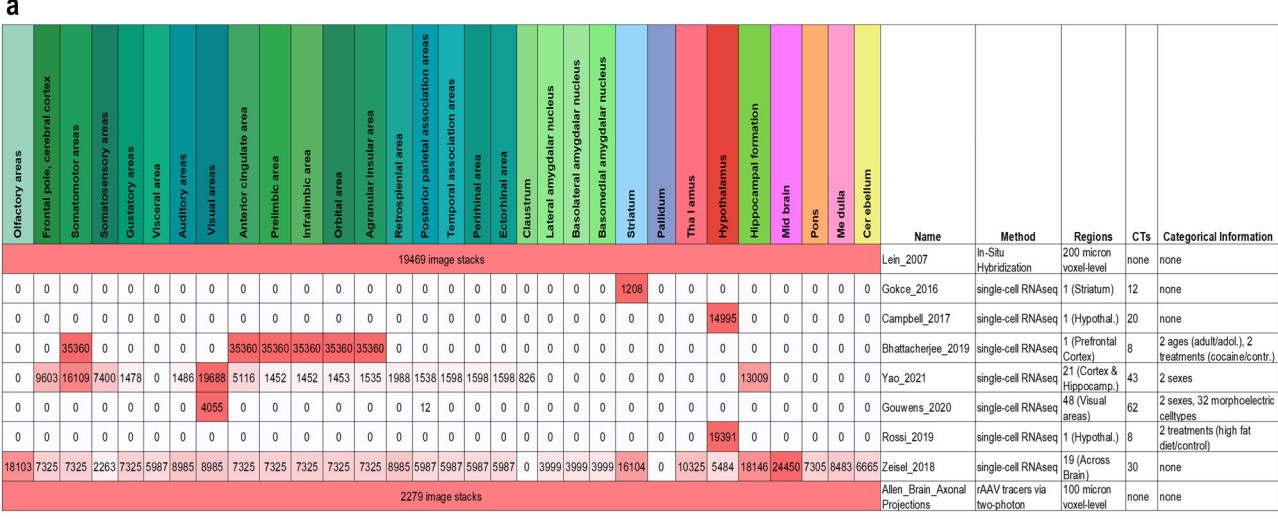

**a**

**b**

Fig. 2 | **Dataset coverage over major anatomical brain regions.** Numbers indicate the sample size/number of images of the datasets in the respective brain regions. Brain region colors represent the used hierarchical brain ontologies from the Allen Institute. **a** Mouse datasets. **b** Human datasets.

RNA sequencing datasets were leveraged using TPM (transcripts per million) for bulk RNA sequencing, and CPM (counts per million) for single-cell/nucleus RNA sequencing to ensure intra-dataset comparability of gene expression[53], at the expense of inter-dataset comparability, which cannot be assumed due to technical biases[54] as well as different experimental conditions and/or sequencing protocols[53]. To circumvent this issue, two steps were taken. First, we limited the comparison to samples from adult subjects to avoid confounding factors due to varying developmental stages[9].

Second, inter-dataset comparability was assessed on rank level, i.e., whether the general order of genes by their TPM/CPM was consistent across datasets. Therefore, we visualized heatmaps that show Spearman rank-based correlation over all genes of mouse and human datasets that cover matching brain regions and cell types (e.g. Battacherjee at al.[34] and Yao et al.[36]), as seen in Fig. 3. Black boxes in the heatmaps mark correlations of matching cell types in different datasets, indicating consistent (ranking of) gene expression. Hence, comparability of relative gene expression between

datasets can be assumed. While the comparison of co-expressed genes via cell type and region specificity would provide further mechanistic insights about gene regulation, this is in general limited because of the absence of matching references (i.e. region specificity would require datasets covering the same brain regions, cell type specificity the same cell types).

For details about the mapping of the datasets, as well as the pre-processing and normalization, see the Methods (Section Data Mapping and Querying and Data Preprocessing and Normalization). Supplementary Data 2 and Supplementary Data 4 provide additional information and code.

## Mapping to a common reference space

The joint exploration of spatial datasets from different resources requires the data to be aligned to a common space[8]. This space acts as a reference, so that spatial locations, such as coordinates or brain region annotations, have the same meaning across datasets. In neuroscience, commonly used reference spaces are typically defined by an anatomical reference template[29,30], a set of

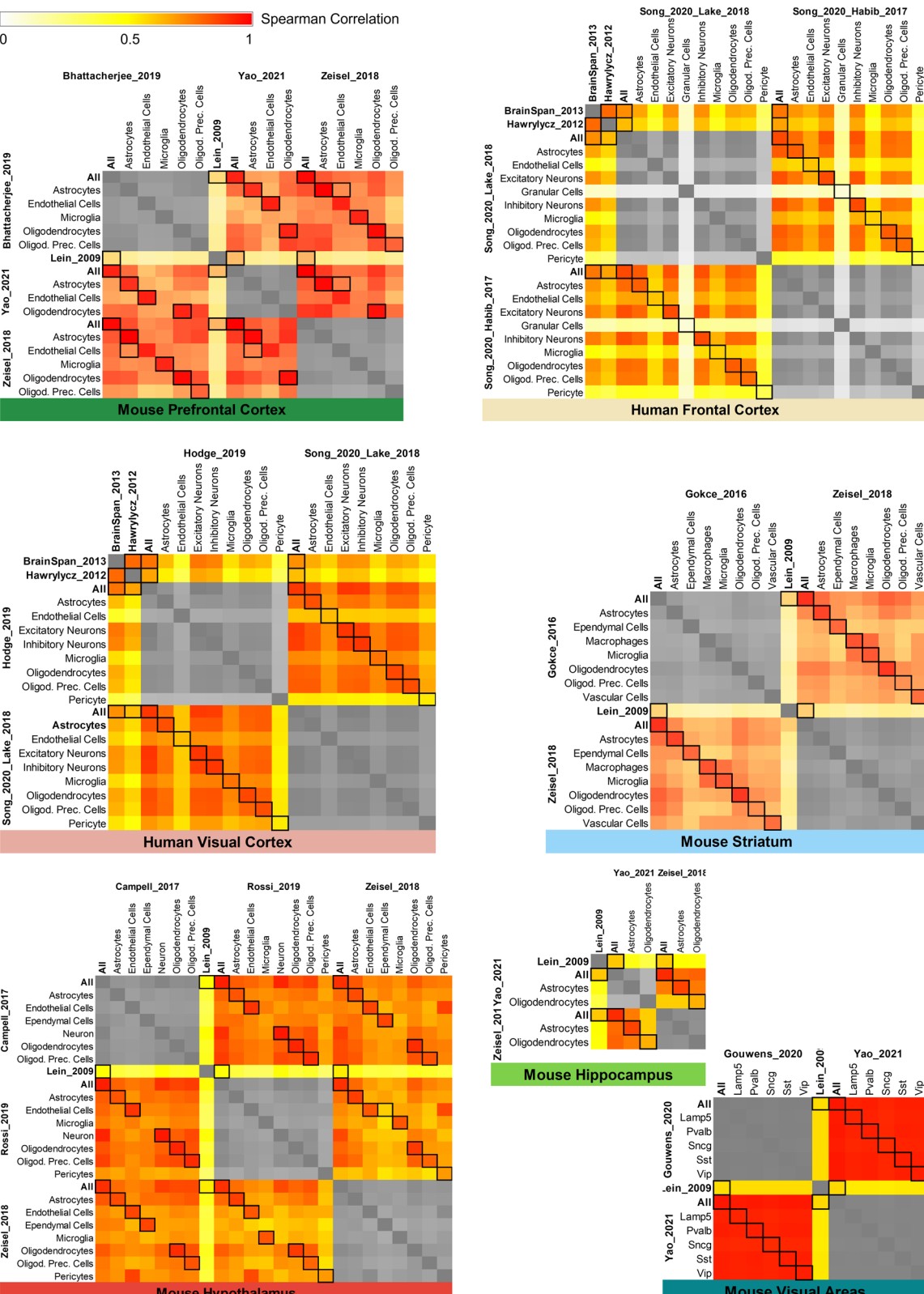

**Fig. 3 | Heatmaps showing the (Spearman rank-based) correlation over all genes of mouse and human datasets that cover the same brain regions and cell types.** Black boxes mark correlations of the same cell types in different datasets. On average, their correlation is higher than for not-matching cell types, which indicates that (the ranking of) gene expression is consistent across datasets (one-sided Wilcoxon test, all p-values≤0.05, except mouse visual areas). Neuronal subtypes in mouse visual areas were already so similar within datasets (all correlation≥0.95), that there was no significant difference across cell types.

structural images that have been combined (e.g. via image registration) to a complete representation of the brain for a group of specimen or a species.

Imaging data, i.e., data that is divided by a 2D/3D grid into pixels/voxels that represent measurements at their respective positions, can then be aligned onto a template by image registration. This involves image transformation and warping to establish voxel-level correspondence (Fig. 1). As templates we used the Allen Mouse Brain Coordinate Framework[29] (0.1 mm resolution) and the ICBM 152 MNI space template[30] (1 mm resolution) based on their widespread use and availability[55]. In principle, there is no limitation to specific templates.

For spatial data that is not derived from imaging, i. e., measurements that have only been generated for specific brain regions, a different approach is needed. Here, we utilized hierarchical ontologies of brain regions, a formal representation of knowledge about species-specific brain anatomy[56] (Fig. 1), i.e., which brain regions it is composed of and how these brain regions are subdivided (hierarchically). The Allen Institute provides ontologies for both mouse and human[29,57,58] that include a mapping onto our respective reference spaces, i.e., a direct correspondence between each coordinate of the reference space to a brain region in the ontology. Since datasets are not necessarily annotated with the same ontology and on the same hierarchical level, they cannot be compared across anatomical scales and resolution directly. Hence, we matched brain region annotations to the corresponding brain regions in the ontology. An outline of this process is shown in Fig. 1, details can be found in the Methods, Section Data Mapping and Querying. Note that these mappings are made explicit in our resource's user interface to ensure transparency, and as a consequence, allow for quality control.

The distributed nature of brain functions across brain networks and gene sets necessitated a thorough exploration of spatial gene expression in the context of brain structural and/or functional connectivity. To this end, we integrated high-resolution imaging data of structural connectivity (for mouse)[24] and resting-state functional connectivity (for human)[7]. Structural connectivity, which describes how brain areas are physically connected via axonal projections, was originally imaged at a 100-μm resolution[24]. The human resting-state functional connectivity, on the other hand, describes brain regions that are linked by correlated activity. This data, sourced from the WU-Minn Human Connectome Project[7], represents the group-average dense, voxel-level correlation of the resting-state BOLD signal of 820 subjects. We selected both the mouse structural connectivity data and the human resting-state functional connectivity data for their high resolution and compatibility with the common reference spaces (see Section Data Preprocessing and Normalization for more details).

### Interactive access and exploration

BrainTrawler was one of the first iterations of an interactive, web-based visual analytics framework[26]. Originally, it was designed to explore large-scale brain connectivity data, such as structural connectivity[24], and to dissect these connections on gene expression level in the mouse brain. This was achieved by providing a visual analytics workflow to identify which genes are expressed in either the source or target regions of these connections, based on spatially mapped gene expression data of 20.000 genes in the Allen Mouse Brain Atlas[25]. Interactivity was achieved by facilitating spatial indexing on volumetric images[59] for the spatially mapped gene expression data, as well as developing a data structure for real-time aggregation of connectivity data with billions of connections[27].

We expand on this effort to handle large-scale transcriptomic datasets for mouse and human, to not only show where genes are expressed, but also how expression differs between cell types and developmental or physiological conditions. We therefore build a spatial database of RNA sequencing and microarray-based gene expression datasets, including the datasets described in the previous section. This spatial database utilizes spatial indexing for aggregating gene expression of datasets in real time, that were aligned to brain regions/voxels of the reference space. To this end, datasets including their meta data (e.g. cell type annotations, age, phenotype, etc.)

were sorted based on their spatial location in the brain (see Methods, Section Spatial Indexing for details).

The exploration of gene expression related to brain connections works in an analogous manner as previously presented in Ganglberger et al.[26], see Fig. 4: First, the user defines a volume of interest (VOI), which can be either an arbitrary manually defined area (via a brush-drawing tool), or a brain region (Fig. 4a, yellow area). For this VOI, a gene expression query can be performed, which computes the mean expression of all datasets that have been aligned to the reference space within the VOI. Results can be refined via a user-defined filter, i.e., selected meta properties data such as certain cell types, phenotypes, and others (details in the Methods, Section Data Mapping and Querying). This approach offers a simplified representation of complex gene expression patterns within specific cell types and brain regions, enabling an accessible visual analytics approach for comparing key differences without losing essential information. To provide differential gene expression insights in relation to other cell types or the whole brain, cell-type-specificity (i.e. gene expression of a cell type vs other cell types) and region-specificity (i.e. gene expression of a VOI vs the whole brain) queries can also be applied, offering a more comprehensive understanding of the gene expression landscape (see Section Data Mapping and Querying for details).

The result of such a queries are lists of genes with the aggregated gene expression. Figure 4c shows how multiple queries results can then be compared in a parallel coordinate system, which allows filtering multiple gene lists by their gene expression. Each axis in the figure represents the result of a gene expression query, and, as a consequence the level of gene expression in the query regions. Each plot line represents a gene. A selection/filtering of genes (shown in the table in the lower part of Fig. 4c) with specific gene expression patterns can be made drawing brushes on an axis. Since queries of different VOIs can be compared, one can use this on the source and target areas from connectivity data for gene expression dissection. Figure 4a, b shows the aggregated outgoing structural connectivity of the VOI in red. While the yellow VOI in Fig. 4a represents the source, the yellow area in Fig. 4b represents the (strongest) targets of the aggregated connections. A comparison of the gene expression of source and target VOI can be seen in Fig. 4c. Here, the axes labeled in red are results of gene expression queries at the source VOI, green ones at the target VOI, performed for different exemplarily selected datasets and cell types.

The increase in available datasets and their inherent complexity (e.g. samples from multiple cell types, developmental states, etc.) makes it necessary to perform large amount of expression queries to cover all available information for genes of interest. Hence, we extended Brain-Trawler's capability to visualize gene expression of multiple resources jointly by developing a lightweight interface (BrainTrawler LITE). BrainTrawler LITE's basic user interface element is a heatmap of the dataset coverage (Fig. 5a). Each heatmap tile represents the sample size/image quantity distribution of a certain dataset (rows) for a certain brain region (columns), similar to Fig. 2. By clicking on a heatmap tile, these data can be selected for further investigation: Either on a gene set level, by entering a list of genes (Fig. 5b), or on a genome-wide level (Fig. 5c), analogously to a gene expression query. Results can be exported as images or as tabulated text files for later use or for sharing. For more details see Methods, Section Brain-Trawler LITE.

### Relating gene expression and connectivity across species uncovers genes and mechanisms for human functional connectivity

The functional (FC) and structural connectivity (SC) of the brain are viewed as a major determinant of cognitive function across species. Altered connection topology and intensity of brain areas commonly correlate to psychiatric conditions such as the autism spectrum and schizophrenia, suggesting that dysconnectivity might lie at the core of these conditions[60–62]. Alongside, genome-wide association studies have discovered genetic loci and polymorphisms associated with these psychiatric conditions, indicating that the relationship between the connectivity of a given brain area and its

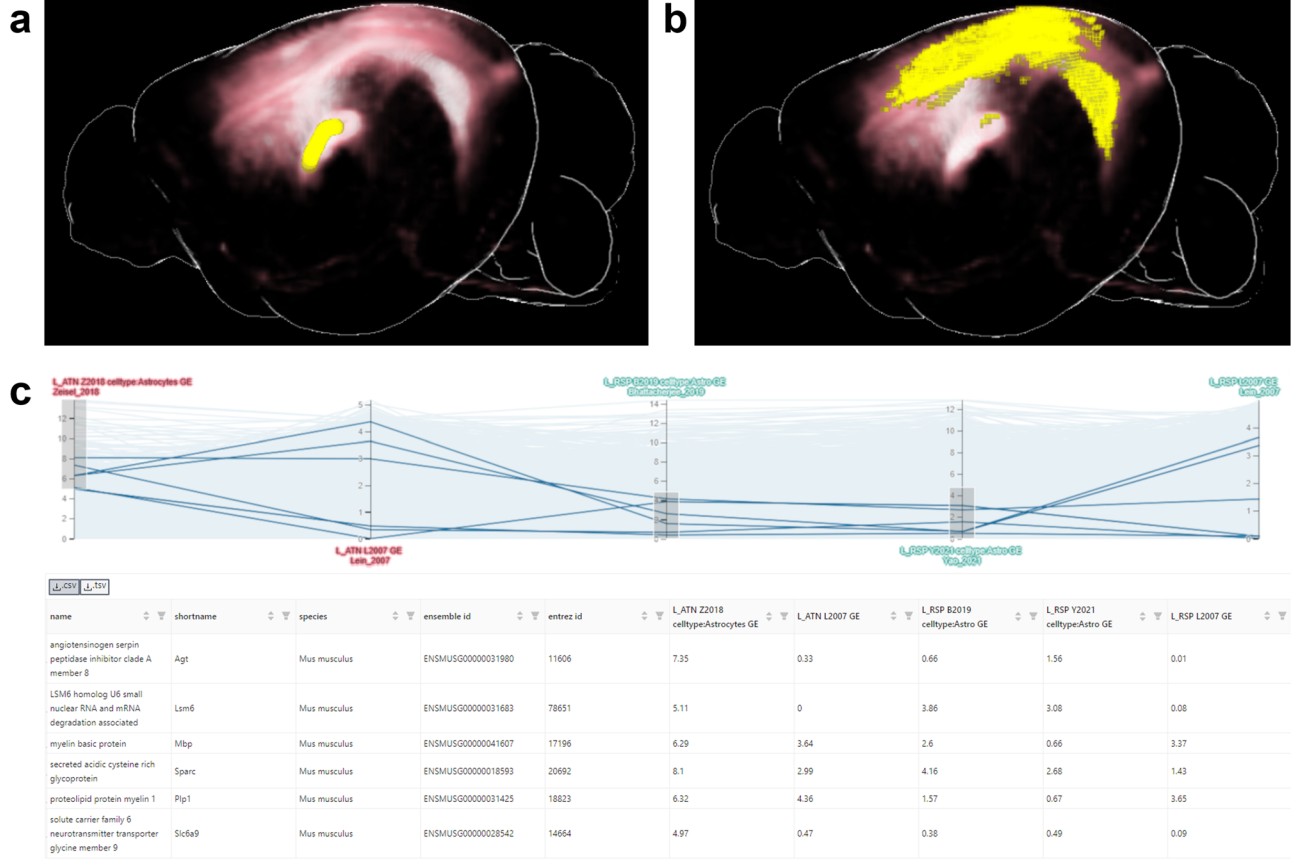

**Fig. 4 | Exemplary gene expression dissection of a structural connection, showcasing the process of defining volumes of interest (VOI), analyzing structural connections, and comparing gene expression patterns. a** Outgoing structural connections (red) from a user-selected part of the Thalamus (yellow), representing the source VOI of the connection. This VOI can be an arbitrary manually defined area or a specific brain region. **b** Target VOI (yellow) of the structural connections (red), representing the strongest targets of the aggregated connections. **c** Gene expression comparison in a parallel coordinates system for source VOI (red axes, one for astrocytes in the Zeisel_2018[37] dataset, one for gene expression in the Lein_2007 dataset) and target VOI (green axes, gene expression in the Lein_2007 dataset, Astrocytes in the Battacherjee_2019[34] and Yao_2020 datasets[36]) using different datasets and cell types. Each horizontal line represents a gene, with its position on the axes indicating the level of gene expression. The axes represent the results of gene expression queries, allowing for comparison and filtering of multiple gene lists. An example subset of genes with low astrocytes expression in the source VOI and high astrocytes expression in the target VOI is selected, demonstrating the potential for gene expression dissection of structural connections.

gene expression likely harbors valuable information on wiring principles of the brain[63,64]. To bridge these domains, the combination of connectomic and gene expression data is a promising approach to discover genetic etiologies and emerging mechanisms that drive regional efferent and afferent connectivity underlying connectopathies, conditions associated with aberrant brain connection topology[65].

The increasing abundance of (cell type specific) gene expression and connectivity data in the mouse is a promising avenue to discover genetic susceptibilities and mechanisms affecting the connectome, with great translational potential. Thus, identifying genetic drivers of FC in humans is of great interest, as it may provide entry points into therapeutic interventions to ameliorate disease burden.

In the context of psychiatry, the insular cortex is of special interest as a core hub in regulating large-scale brain networks in humans[66] and rodents[67]It is also involved in interoceptive, cognitive and affective processes[68,69]. Notably, insular functional dysconnectivity is an indicator of common psychiatric conditions[70,71]. We segmented the insula into its agranular and granular portions, as they are anatomically and functionally distinct[72]. While the posterior granular insula (GI) is a primary sensory area with rich afferents for interoceptive information, the anterior agranular insula (AI) represents an associative area with increasing multimodal integration[68]. Therefore, this system is an ideal model area to discover novel genetic factors shaping the connectivity of cortical areas of distinct architecture (agranular vs granular insula), in a highly relevant translational setting.

First, to allow for optimal cross-species inferences, we selected consensus areas between the rodent and human brain, covering 10 major subcortical regions (Supplementary Data 5). Next, source and target connectivity data with these areas were sampled for the AI (combined "L_Agranular insular area, dorsal part" and "L_Agranular insular area, ventral part") and GI ("L_Visceral area") (Supplementary Fig. 1, left). Because the human Allen Brain Atlas (ABA) does not discern by granularity, human FC data for AI and GI with the consensus areas were sampled by brushing agranular and granular areas of the short and long insular gyri (Supplementary Fig. 1, right, according to[73]. Within-species analysis shows a correlation between source and target SC in the mouse GI, but not in the AI (Fig. 6a). Overall, AI and GI connectivity is correlated in rodents and humans (Supplementary Fig. 1a), although to different extents between sources and targets. As expected, human FC is not significantly correlated to mouse SC, suggesting relevant functional differences between species and/or connection modality (Fig. 6b).

To assess the relationship between gene expression and connectivity, we extracted expression data of major excitatory and inhibitory cell types of the 10 subcortical consensus areas for which expression data is available from Zeisel_2018[37] (Mouse) and Hawrylycz_2012.[20,21] (Human) (Supplementary Data 5). These were correlated with FC and SC within humans and mouse, respectively (see Supplementary Data 6 for Top and Bottom 1% correlated genes). To identify potential basic driver genes for insular connectivity (i.e. those

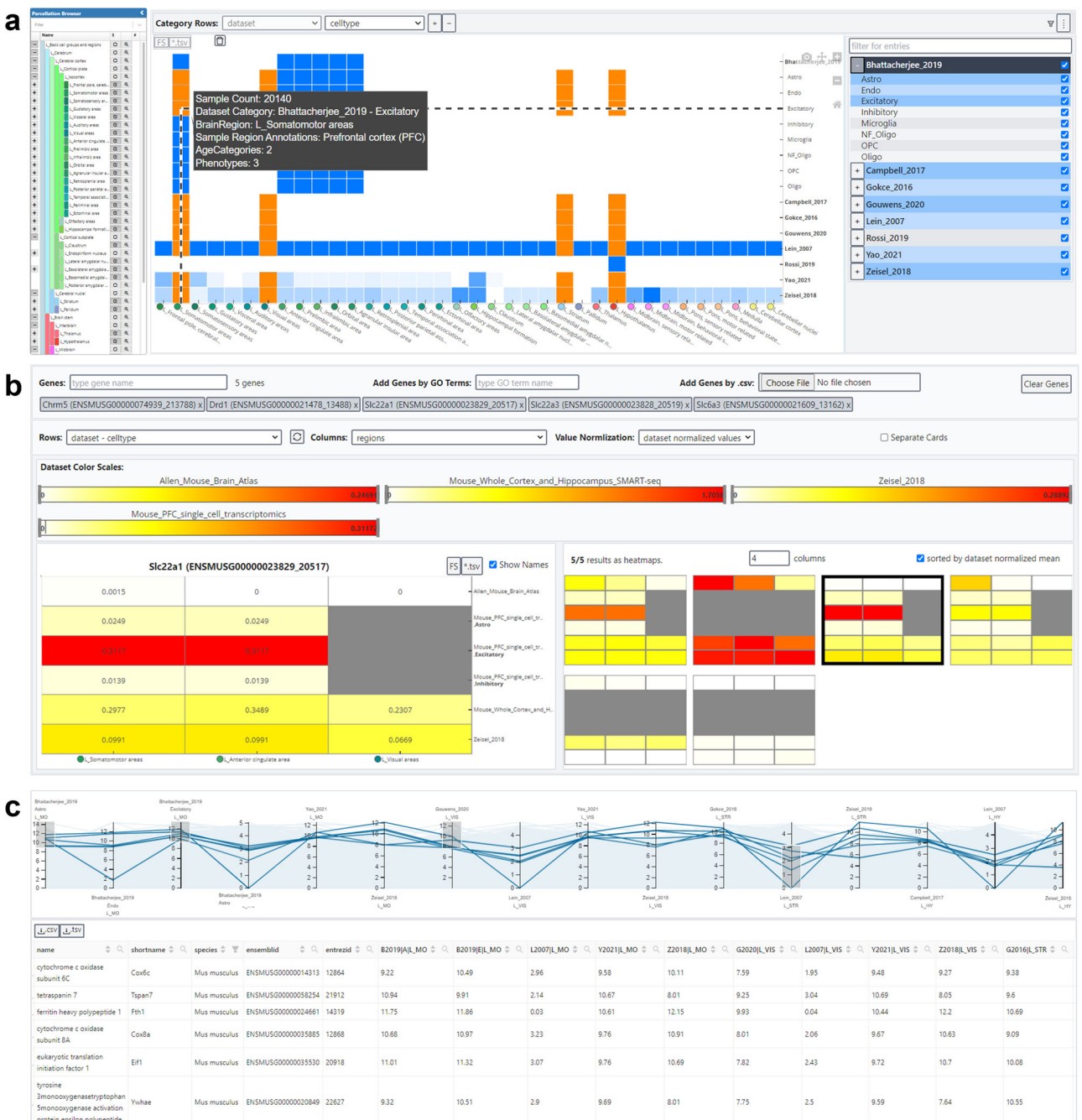

**Fig. 5 | BrainTrawler LITE interface for comparative visualization of gene expression across datasets. a** The dataset coverage heatmap shows the distribution of sample size/image quantity across brain regions (columns) and datasets (rows), subdivided by meta data attributes such as cell types, phenotypes, etc. Brain regions and meta data categories can be adapted via tree-like UI elements on the sides, the tooltip shows the exact composition (sample/image count, meta data categories, etc.) of the respective heatmap tile. Orange tiles shows a selection of data for gene expression visualization (in **b**, **c**). **b** Gene expression heatmaps of five selected genes. Rows and columns represent the selected (orange) tiles in the dataset coverage

heatmap (in **a**). Color scales are separated per dataset (between 0 and the maximum value shown for all selected genes). The right side shows the expression of genes for each dataset separately as small multiples, the left side shows one selected gene with more details (labels, values, etc.). Gray tiles are missing data. **c** Parallel coordinates system showing the gene expression of all genes in the selected dataset on axes, each representing the average expression of genes (blue lines) of the samples/images of each selected (orange) tile in the dataset coverage (in a). Via drawing brushes on the axes, the genes in the parallel coordinates system can be filtered. Filtered genes are shown below in a table.

conserved across the available mouse structural and human functional connectivity data), we determined the overlap between human and mouse datasets. This resulted in a total of 26 genes for AI (Fig. 6c, left) and 21 genes for GI across mouse sources and targets (Fig. 6c, right; see Supplementary Fig. 2 for source- and target-specific analysis and Supplementary Data 7). Association analysis for brain-related categories on these genes in Open Targets [74] suggests that they are involved

in processes relevant to psychiatric conditions (Fig. 6d). In this context, our workflow recovered several genes previously associated with autism (AI: 4 genes; GI: 2 genes) and schizophrenia (AI: 8 genes; GI: 6 genes) (see Supplementary Data 8). Among the positively correlated we find *attractin-like 1* (*ATRNL1*) specifically for AI, a gene previously found to be mutated in a human patient diagnosed with autism[75]. The *estrogen receptor 2* (*ESR2*), which is among few genes with a link to

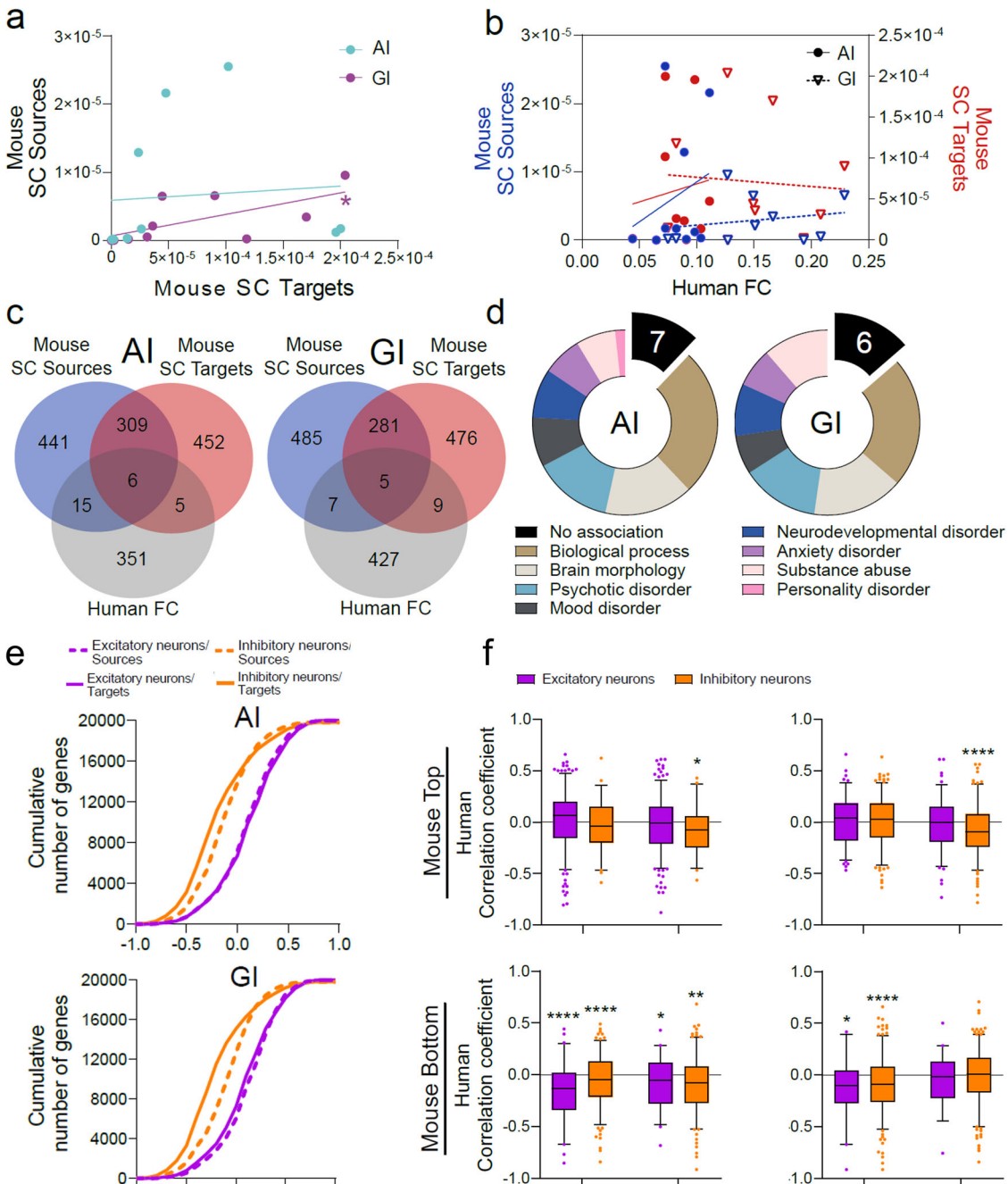

**Fig. 6 | Relating regional gene expression with connectivity across species identifies potential connectivity driver genes, where cell-type-specific analysis reveals a conserved inhibitory mechanism across species and connection modality.**
**a** Mouse source and target connectivity is significantly correlated for GI (Spearman $r = 0.65$, $p$-value = 0.04), but not for AI ($r = 0.08$, $p$-value = 0.82). **b** Mouse SC source and target connectivity are not significantly correlated to Human FC (Spearman; AI: source $r = 0.22$, $p$-value = 0.54, target $r = 0.22$, $p$-value = 0.54; GI: source $r = 0.22$, $p$-value = 0.54, target $r = -0.05$, $p$-value = 0.89). **c** Overlap of respective top and bottom 1% of genes correlated to connectivity across species (Mouse SC source/ target and Human FC). **d** Brain-related associations of overlapping genes in c (total of 26 genes for AI, 21 genes for GI; see Supplementary Data 7 for summary). **e** Cumulative distribution of correlation coefficients resulting from the correlation of cell-type-specific gene expression and Mouse SC (sources/targets) across excitatory and inhibitory cell types for AI and GI. Kolmogorov–Smirnoff tests revealed significant differences for AI and GI ($p$-value ≤0.0001) between excitatory and inhibitory cell types for within source and target connectivity, respectively. In addition, significant differences ($p$-value ≤0.0001) between source and target connectivity within excitatory and inhibitory neurons, respectively, were found for both AI and GI. **f** Selection of correlation coefficients of human genes with homologs in the top/ bottom 1% correlated genes in the mouse. Analysis is based on direction of correlation in the mouse dataset (top: Top 1%, bottom: Bottom 1%), mouse SC type (left: Sources, right: Targets), cell type of origin (color) and connected region (AI, GI). Significance was tested by one-sample t-test against zero (chance level). *$p$-value ≤ 0.05, **$p$-value ≤ 0.01, ****$p$-value ≤ 0.0001.

schizophrenia, was found to have switched from a strong positive correlation for AI (sources) in mouse to a strong negative correlation in humans (Supplementary Data 8). We also detected several genes previously not linked to brain-related disorders, potentially

identifying novel genetic factors that contribute to dysconnectivity phenotypes.

It is established that brain oscillations are governed by recurrent inhibitory networks[76], suggesting that genes expressed in inhibitory neurons

might drive FC. To address this systematically, we harnessed the available cell-type-specific gene expression data in the mouse as an entry point. We extracted gene expression data of excitatory and inhibitory cell types from the 10 consensus areas in the mouse using the "Cell type specificity" query (Supplementary Data 5). This approach emphasizes cell-type-specific genes and thus enhances contrast between cell types. We then correlated this expression data to the mouse SC of AI, and GI (sources and targets) to the consensus areas. Interestingly, we found that there is a shift of correlation coefficients obtained towards negative correlations in genes expressed in inhibitory neurons (Fig. 6e). This suggests that inhibitory mechanisms by inhibitory neurons might shape mouse SC.

We then tested their negative correlation in human FC. This is most dominant for genes negatively correlating with mouse source connectivity (Fig. 6f, top left), and specific to AI for genes from mouse target connectivity (Fig. 6f, top right). Interestingly, for mouse SC positively correlated genes, the direction of correlation is inverted and specific to GI/inhibitory neurons. This may indicate important differences between species and/or connection modality (Fig. 6f, bottom). Among the strongest candidates we found is *TNF superfamily member 12* (*TNFSF12*), a gene dysregulated in patients diagnosed with various psychiatric disorders[77–79].

In summary, despite the fact that Human FC and mouse SC are not correlated (Fig. 6b), this workflow uncovered some preserved relationship between gene expression and connectivity across species and connection modality. Therefore, this workflow may allow to further mine data for (cell type specific) genes and mechanisms driving FC in humans.

## Discussion

We created a discovery framework that utilizes data from current popular large-scale genetic and brain network initiatives to rapidly screen for neural circuitry underlying specific brain functions, behaviors, or psychiatric symptoms at comparably low computing costs. Our platform has several implications for both basic and biomedical research, and may directly impact subsequent, circuit-genetic experiments such as electrophysiology, opto-, and pharmacogenetics. When performed at large scale with behavior-specific genes, our approach has the potential to refine knowledge about the functional organization of the brain beyond simple anatomical domains. Importantly, our methodology for generating and exploiting data resources could be applied to other model organisms for which spatially mapped gene expression, network, and genetic information is, or will be, available, for example fruit flies or zebrafish. Using our platform, we showcase this by uncovering a preserved relationship between gene expression and connectivity across species and connection modality. We explored excitatory and inhibitory cell types, and identified several genes that were previously not linked to brain-related disorders, potentially identifying novel cell-type specific factors that contribute to dysconnectivity phenotypes.

Inevitably, BrainTACO has some limitations. First of all, in its current state it does not cover the full brain on a single-cell/nucleus level. To provide a resource as versatile as possible, we focused on covering the brain at least the level of most major anatomical brain regions, so that neuroscientists will likely find data related to their own research focus. This was not entirely possible for human datasets (e.g. Amygdala, Thalamus, Hypothalamus) due to a lack of studies covering these areas. In principle, BrainTACO is not static, and can be extended with new datasets. Future studies, as well as further improvements of technologies, such as spatial transcriptomics, will help to close this gap.

Another limitation is that it is in general not reasonable to compare (absolute) gene expression values, even normalized ones (e.g. TPM, CPM) across datasets[53,54], especially with respect to batch effects. We circumvent this issue by providing dataset-specific color scales for gene expression heatmaps, and advise to compare absolute gene expression only relative to other genes within the same dataset. As an alternative, we provide gene ranks for gene expression queries, i.e., their relative position in a list of genes sorted by their expression. This relative measure, in terms of how is a gene's expression ranked relative to all other genes for a certain cell type/brain region, acts as a normalization to adjust for batch

effects and other sources of technical variation, but may not completely eliminate them.

Furthermore, the computed mean expression, region and cell type specificity, or enrichment scores do not include information about the spread of the data, which is typical for traditional analytical approaches, such as t-SNE plots. Hence, it is not known how homogenous the expression is across the selected datasets, for example for a certain cell type. This limitation of analytical power greatly increases the ease of use of the platform, a cornerstone of BrainTrawler, whose mission is to enable computationally agnostic neuroscientists to run complex analysis. Nevertheless, an integration of more sophisticated, in-depth analyses methods might be considered for future releases.

The user interface provided sufficient visualization modes for the scope of our case studies, i.e. the number of queries, datasets and genes was well within BrainTrawler's capabilities. In general, there is no limit regarding how many genes, datasets or query results can be visualized, but outside typical analytical workflows, such as our case study, there are some practical limitations within our user interface: The parallel coordinates system does not scale well to more than twenty queries, due to space constraints in a typical browser window. The same is true for visualizing the expression of hundreds of genes in BrainTrawler LITE's gene expression heatmaps, since the overall context might be lost when the visualization does not fit on one screen and scrolling is needed.

Overall, the integration of heterogeneous gene expression and connectivity data from mouse and human into BrainTrawler is a powerful resource for hypothesis building in the field of behavioral/functional neuroscience and for drug target identification. Its coverage of RNA sequencing data, especially on a single-cell/nucleus level for the majority of brain regions, only limited by the public availability of the data, enhances BrainTrawler's capabilities of investigating molecular mechanisms. By making the BrainTACO resource available via a web-based visual analytics workflows, we enable quick access without manual data aggregation via scripting, and consequently without needing the expertise of a bioinformatician. Future integration of novel spatial datatypes, such as spatial transcriptomics, have the potential to make this resource even more versatile.

## Methods
### Data preprocessing and normalization
**Single-cell/nucleus RNA sequencing data.** We integrated 21 single-cell/nucleus RNA sequencing datasets in total, 7 mouse datasets, and 14 human. Out of the 14 human datasets, 12 are from the Song et al.[9] meta dataset STAB available from http://stab.comp-sysbio.org. STAB consists of 13 datasets[38,42–52], of which we omitted the dataset by Hodge et al.[38] and downloaded the data from Hodge et al.'s original resource (http://celltypes.brain-map.org/rnaseq), since it had been extended by several brain regions after STAB's submission (primary motor cortex, primary somatosensory cortex, and primary auditory cortex). The other single-cell/nucleus RNA sequencing datasets were obtained via the data availability statements in their referenced publications.

STAB datasets were not further preprocessed, filtered or normalized, since this has already been done consistently across all 12 datasets[9]. The remaining single-cell/nucleus RNA sequencing datasets were pre-processed with the Seurat (v4.1.0) R package[80] to remove batch effects. Low-quality cells, empty droplets, and doublets were removed by filtering out low (less then 50) or high (more than 5000) unique gene counts, or if their unique gene counts were identified as outliers (five times lower or higher than the mean absolute deviation from the median). Final cell counts can be seen in Table 1 (Filtered Cell Counts). Note that most available datasets were already filtered by similar criteria, which explains the high similarity of original and filtered cell counts. Cells that could not be matched to brain regions were removed, which was only the case for samples from the medial ganglionic eminence (transient structure in the developmental brain) in the Nowakowski et al.[42] data from STAB.

Genes without cell counts across the datasets where removed, since they do not show biological variability. Genes were then matched by gene

**Table 1 | Original Cell Counts of the retrieved datasets, filtered cell counts after preprocessing, and genes matched to BrainTrawler's gene database**

| Dataset | Species | Original Cell Counts | Filtered Cells Counts | Matched Genes |
|---|---|---|---|---|
| Gokce_2016 | Mouse | 1208 | 1208 | 17077 |
| Campbell_2017 | Mouse | 20921 | 14995 | 29579 |
| Rossi_2019 | Mouse | 20194 | 19391 | 23613 |
| Bhattacherjee_2019 | Mouse | 35360 | 35360 | 15720 |
| Yao_2021 | Mouse | 74974 | 74676 | 36549 |
| Gouwens_2020 | Mouse | 4270 | 4067 | 34846 |
| Zeisel_2018 | Mouse | 160796 | 135626 | 22752 |
| Lee_2020 | Human | 125468 | 70718 | 30191 |
| Hodge_2019 | Human | 49417 | 47432 | 40180 |
| Song_2020_Nowakowski_2017 | Human | 4261 | 921 | 23830 |
| Song_2020_Darmanis_2015 | Human | 466 | 416 | 23830 |
| Song_2020_Zhong_2018 | Human | 2394 | 2005 | 23830 |
| Song_2020_Fan_2018 | Human | 4664 | 3916 | 23830 |
| Song_2020_Li_2018_Part1 | Human | 1512 | 701 | 23830 |
| Song_2020_Li_2018_Part2 | Human | 17093 | 16840 | 23830 |
| Song_2020_Lake_2017 | Human | 36166 | 33862 | 23830 |
| Song_2020_La_Manno_2016 | Human | 1977 | 1869 | 23830 |
| Song_2020_Habib_2017 | Human | 11859 | 10747 | 23830 |
| Song_2020_Welch_2019 | Human | 40453 | 39447 | 23830 |
| Song_2020_Liu_2016 | Human | 276 | 252 | 23830 |
| Song_2020_Onorati_2016 | Human | 1608 | 476 | 23830 |

symbol, Ensembl ID or Entrez ID to BrainTrawler's gene database, which was obtained via the Genome wide annotation for mouse[81] and human[81] via the bioconductor package. The amount of matches can be seen in Table 1. For each gene, expression levels were normalized by computing CPM (counts per million) to ensure intra-dataset comparability of gene expression[53]. For better readability/interpretability, CPM was log2 normalized (using an offset of 1 to account for zeros).

**Bulk RNA-sequencing data**. To fill gaps in subcortical single-cell/nucleus RNA sequencing data for the human, we integrated two bulk RNA sequencing datasets from the GTEx and BrainSpan consortia[40,41]. GTEx data was downloaded from the GTEx portal (https://gtexportal.or) in version 8 as gene TPM (transcripts per million). BrainSpan data was obtained from the BrainSpan portal (https://www.brainspan.org/) as normalized RPKM (reads per kilobase of transcript) expression values, and converted to TPM according to Zhao et al.[53]. We applied log2 normalization (using an offset of 1 to account for zeros) to both datasets, similar to single-cell/nucleus RNA sequencing data processing.

**Microarray gene expression data**. Microarray gene expression data was retrieved from the Allen Human Brain Atlas by Hawrylycz et al.[20,21] via the Allen Brain Atlas API. This data assembled gene expressions from 3702 samples of six donors, labeled with their according brain region in the ontology provided by the Allen Institute[57], which ensures equivalent scaling across donors. We normalized gene expression values based on an outlier-robust sigmoid function, before rescaling the normalized values to a unit interval (0-1), as suggested by Arnatkevičiūtė et al.[82].

**In situ hybridization data**. Whole-brain gene expression in-situ hybridization data was retrieved from the Allen Brain Atlas API as volumetric images for 19479 genes. To create these volumetric images, the Allen Brain Atlas divided the in-situ hybridization slice images on cellular resolution into a 200-µm resolution grid. For each grid division, expression energy was computed, i.e. the sum of the expression intensity

of all pixels within the division, divided by the sum of pixels within the division[83]. The expression energy for all grid divisions can then be seen as a 200-µm resolution volumetric image. To make this data available in BrainTrawler, we log2-normalized the data and encoded them as 8 bit volumes, with a size of 155KB each (~3GB in total).

**Structural connectivity data**. Structural connectivity was generated similar to previous publications[26,27,84]. Here, the connectivity was retrieved from the Allen Brain Atlas API as volumetric images, showing structural connectivity of 2173 injection sites to their target sites[24]. These 2173 images were generated on a 100-µm resolution by labeled rAAV tracers via serial two-photon tomogaphy[24]. For each image, the injection site is given by coordinates in the reference space defined by the Allen Mouse Brain Coordinate Framework[29], and an injection volume, depicting the volume around the injection site affected by the tracer. Hence, the connectivity for an injection site is defined by all voxels within its injection volume. For every voxel in the reference space, we took the connectivity from the covering injection volume. If a voxel was covered by multiple injection volumes, and therefore by multiple injection sites, we combined them by taking the maximum connectivity for each target. To compensate for low count of injection sites on the left hemisphere, we mirrored the connectivity, effectively doubling the original 2173 injection sites to 4346. To minimize the amount of false positive connections, the data was thresholded by values $<10^{-4.5}$ according to Oh et al.[24], Extended Data Figure 7. The result was a dense $\sim 67{,}500 \times 500{,}000$ structural connectome (~67500 source voxels, covering injection volumes with ~ 500,000 target voxels within the mouse brain), with ~90GB stored in a csv format.

**Resting-state functional connectivity data**. Resting-state functional connectivity data was downloaded from the WU-Minn Human Connectome Project[7] via the ConnectomeDB (https://db.humanconnectome.org/). We specifically chose the human resting-state functional connectivity data over the structural connectivity diffusion tensor imaging data provided by the Human Connectome Project[7],

**Fig. 7 | Mapping of exemplary RNA sequencing data to, and retrieving from a common reference space. a** Samples of the Thalamus (green) and Hypothalamus (brown) of an exemplary RNA sequencing count matrix are mapped manually to a brain regions in an hierarchical ontology via literature research. Since mapping of the ontology to the reference space is known, samples can be mapped to individual voxels of the reference space, and hence to every anatomical level in the ontology. **b** Aggregating the average gene expression for all samples from a coarser anatomical level (Diencephalon) than the original annotations (Thalamus and Hypothalamus). **c** Aggregating the average gene expression for all samples from a equal or finer anatomical level (Thalamus or Dorsal Thalamus) than the original annotations (Thalamus) leads to the same results.

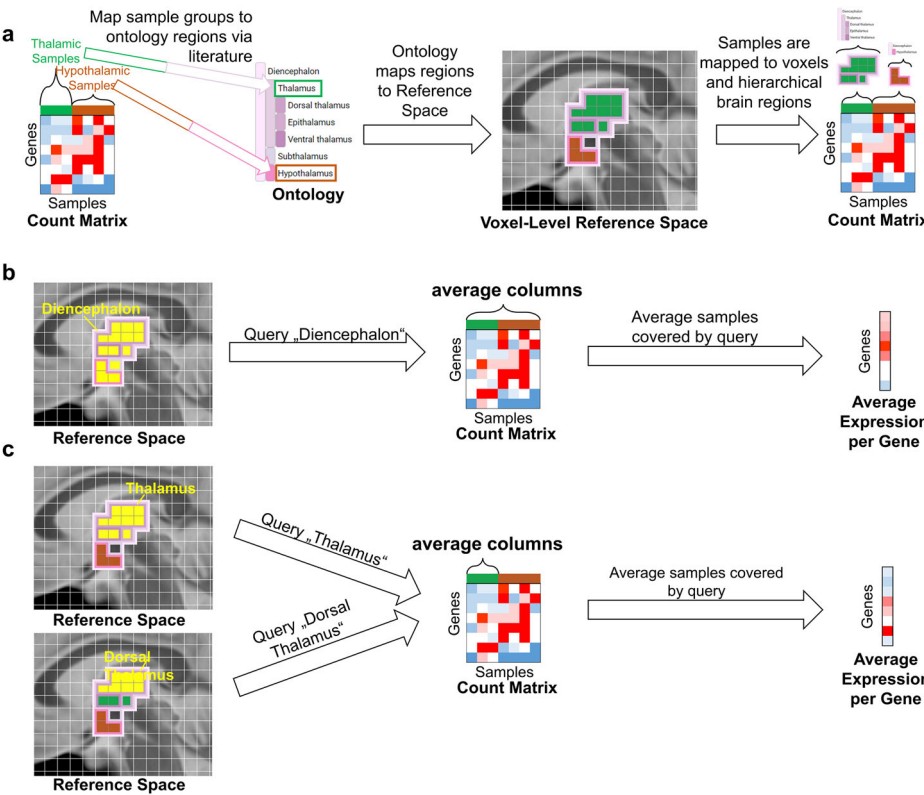

because it was more readily preprocessed and provided higher resolution than region-level[85], better showcasing the capability of BrainTrawler to support even voxel-level connectivity. The data was available as an average functional connectivity matrix of 820 subjects, given as a dense ~90,000 × 90,000 functional connectome in "grayordinate" space[7], where a grayordinate is either a voxel (subcortical gray matter) or a surface vertex (cerebral cortex). The Human Connectome Project provides the cortical surface (vertices) of the 152 MNI template[30] in grayordinate space. We used the Connectome Workbench platform[7] to extract the surface vertices coordinates in MNI space (see script getVoxelAndVerticesToMNISpaceMapping.m provided in Supplementary Data 1). To finally map the functional connectivity matrix to our 1mm resolution ICBM 152 MNI space template reference space, we assigned every voxel of the template with the closest cortical surface vertex coordinate or subcortical voxel coordinate. This resulted in a dense ~87,000 × 87000 functional connectome, with ~45*GB*, stored as csv.

## Data mapping and querying

Imaging data shown in this paper, namely in-situ hybridization data[25], axonal projection connectivity[24] and resting state functional connectivity[7] was already aligned to the reference spaces used for this resource[29,57,58]. Novel datasets can be aligned via tools such as the QUINT workflow[86] or the ANTS frame work[87]. Data that does not meet the resolution of the reference space, can be up- or downsampled via nearest-neighbor interpolation. This was the case for the in-situ hybridization data, which had a lower resolution (200 μm) than the Allen Mouse Brain Coordinate Framework[29] (100-μm resolution).

Region-level data, such as microarray gene expression data and RNA sequencing data are typically encoding gene expression as count matrix[88], reporting the frequency of gene transcripts for samples. For dataset included in this study, these samples originate from brain regions. We manually mapped these brain regions to the corresponding brain regions of our reference ontology (Allen Brain Institute Atlases), based on the region name and description in the dataset's reference publications. To ensure transparency, and hence quality control, the detailed mappings are available in

the resource's user interface (Browse Database, then select a dataset to see details such as the dataset's mapping), and in the supplemental material.

The process of mapping region-level data to, and retrieving it from a reference space is outlined exemplarily in Fig. 7. Code for the mapping can be found in Supplementary Data 1. In this example, data are samples from the Thalamus and Hypothalamus (Fig. 7a). Since the ontology maps to the corresponding voxels of the reference space, each voxel can be related to samples that originated from the voxel's brain region. The hierarchical nature of the ontology enables the querying of gene expression on multiple anatomical levels. For example, querying the average gene expression in the Diencephalon, the parent region of Thalamus and Hypothalamus, will aggregate over all samples of the count matrix (Fig. 7b), while a query on the Thalamus or a subregion of the Thalamus (e.g. Dorsal Thalamus) will result in an aggregation over the thalamic samples (Fig. 7c). We want to point out, that thalamic samples do not necessarily represent dorsal thalamic samples, hence we make the samples origin explicit in our resource's user interface (Fig. 5a, "Sample Region Annotations").

We implemented four different variants of gene expression queries to cover different use cases, such as region-specificity or enrichment. These queries were defined on a gene expression matrix of dataset *d* as

$$\mathbf{M}^d = (m^d_{g,s})_{g \in \mathbf{G}, s \in \mathbf{S}}, \mathbf{M}^d \in \mathbb{R}^{|\mathbf{G}| \times |\mathbf{S}|} \tag{1}$$

where each row represents a gene $g \in \mathbf{G}$ and each column a sample (or a voxel in case of imaging data) $s \in \mathbf{S}$. $\mathbf{V} \subseteq \mathbf{S}$ represent all samples within the VOI, $\mathbf{C} \subseteq \mathbf{S}$ samples of a certain cell type, and $\mathbf{F} \subseteq \mathbf{S}$ a samples filtered by meta data other than cell types.

**Mean expression query.** Computing the mean gene expression within the VOI for each gene $g$

$$meanexpression(g) = \frac{1}{|\mathbf{V} \cap \mathbf{C} \cap \mathbf{F}|} \sum_{s \in (\mathbf{V} \cap \mathbf{C} \cap \mathbf{F})} m_{g,s} \tag{2}$$

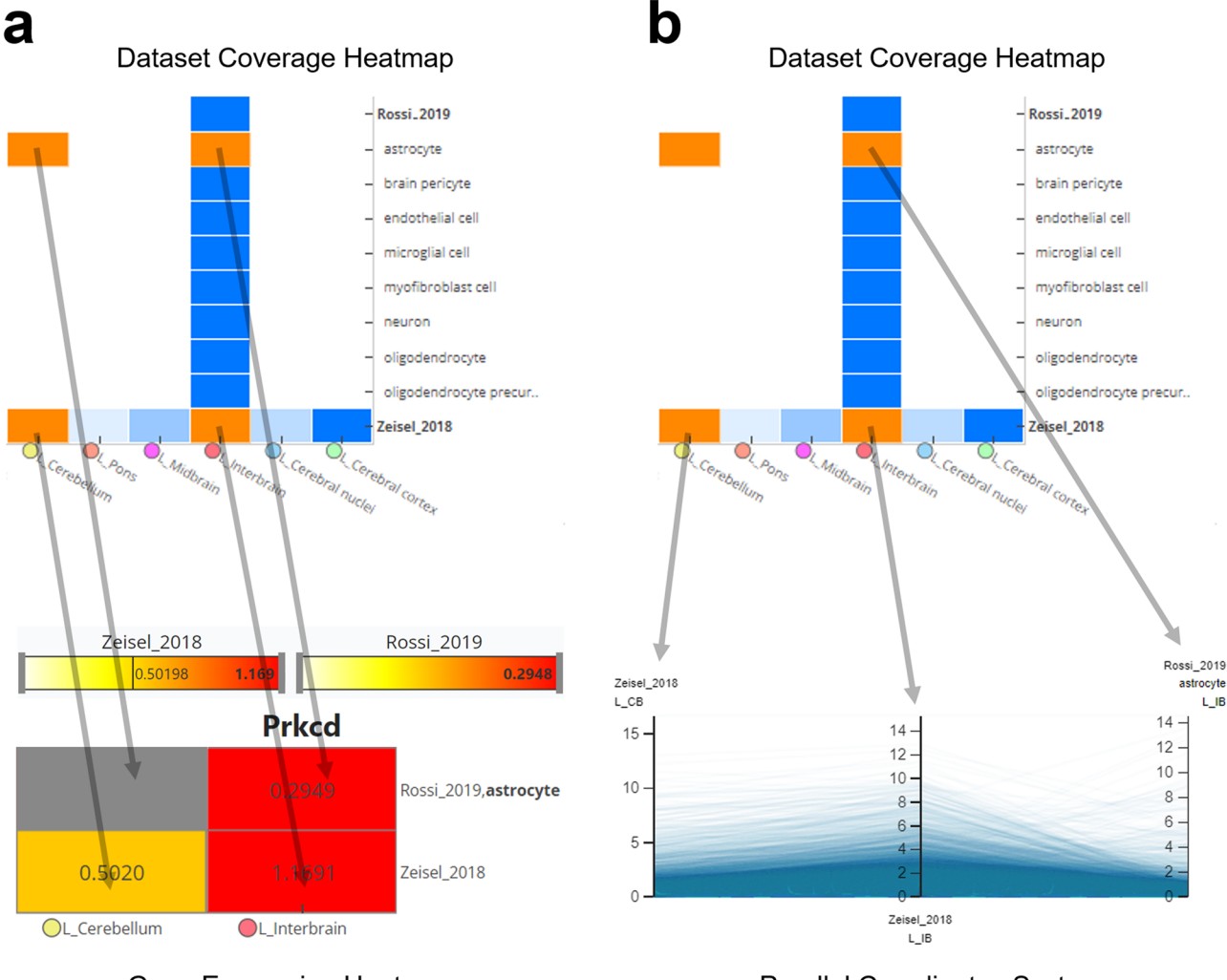

**Fig. 8 | Concept of how a selection in the dataset coverage heatmap transfers to gene expression heatmaps and the parallel coordinates system.** Each tile represents a subset of the resource, i.e., samples/images of a certain brain region of a certain dataset (and of a certain meta data category). Each selected tile (orange) has a direct representation as tile in the gene expression heatmap (**a**) and as axis in the parallel coordinates system (**b**). The values in (**a**) and (**b**) are the averaged expression values (e.g. CPM, TPM, etc.) over all samples/images represented by the selected tile. Tiles without samples (missing data) are rendered gray in (**a**), or omitted for (**b**).

**Region specificity query**. To account for regional specificity, we compute for each gene $g$ the mean gene expression within the VOI, and normalize it to the mean expression of the rest of the brain:

$$regionspecificity(g) = \frac{\frac{1}{|V \cap C \cap F|} \sum_{s \in (V \cap C \cap F)} m_{g,s}}{\frac{1}{|C \cap F|} \sum_{s \in (C \cap F)} m_{g,s}} \quad (3)$$

**Cell type specificity query**. This query can be used to see how specific the expression of a certain cell type is. In this case, for each gene $g$, the mean gene expression within the VOI is computed for all samples of a certain cell type $C \subseteq S$, and normalized by the expression over samples of all cell types within the VOI:

$$celltypespecificity(g) = \frac{\frac{1}{|V \cap C \cap F|} \sum_{s \in (V \cap C \cap F)} m_{g,s}}{\frac{1}{|V \cap F|} \sum_{s \in (V \cap F)} m_{g,s}} \quad (4)$$

**Enrichment query**. This query can be used to see how specific the expression is for cell types $C \subseteq S$ or different meta data $F \subseteq S$. In this case, the mean gene expression within the VOI is computed for all samples of the selected filter and cell type, and normalized by the expression over all samples within the VOI:

$$enrichment(g) = \frac{\frac{1}{|V \cap C \cap F|} \sum_{s \in (V \cap C \cap F)} m_{g,s}}{\frac{1}{|V|} \sum_{s \in (V)} m_{g,s}} \quad (5)$$

**BrainTrawler LITE**

The basis of BrainTrawler LITE is the dataset coverage heatmap, showing the distribution of samples/images across brain regions (columns) and datasets (rows), subdivided by metadata categories such as cell types, phenotypes, etc. (Fig. 5a). Regions can be dynamically set via a tree-like structure showing the brain ontology (Fig. 5a, left). Datasets can be subdivided by their metadata categories (e.g. split by cell types, Fig. 5a, right), so that the rows do not only represent datasets, but also subsets thereof, e.g. a row for each cell type per dataset. Hovering over the individual tiles of the heatmap reveals a summary of the data for the respective brain region and dataset (or subset), for example sample count and the original region annotation of the data. The original region annotation is especially relevant to identify the data's origin, and hence, the data's potential relevance for the user.

In our first example (Fig. 5b), one gene expression heatmap is generated for each of the entered genes. Here, a gene expression heatmap shows

the averaged expression for all samples/images covered by each of the selected tiles in the dataset coverage heatmap. This means, that if the user selects a tile in the dataset coverage heatmap of a certain brain region, and a certain cell type of a certain dataset, each gene expression heatmap will contain the same tile, showing the averaged expression of all the tile's covered samples/images (Fig. 8a). To deal with gene lists with dozens of genes, we used a small multiples visualization[28], (Fig. 5b, right) so one can visually identify patterns while maintaining an overview. Clicking on individual gene heatmaps will show a detailed view on the left-hand side (Fig. 5c, left) displaying the exact expression values and row/column labels. The coloring is set by individual color scales per dataset (same color, but the range depends on the datasets maximum value), since, as already mentioned before, values are not directly comparable across datasets (Fig. 5c).

In the case of investigation on a genome level (Fig. 5c), a parallel coordinates system is used analogue to the dissection of connections shown in Fig. 4. Here, each plot line represents a gene, indicating the averaged expression along axes for each selected tile (Fig. 8b). Genes can be filtered by brushing along an axis (Fig. 5c) to find genes with specific gene expression patterns.

### Spatial Indexing
Real-time queries on the resource's data was achieved by spatial indexing, depending on the datatype:

- *Connectivity data*: For real-time aggregation of connectivity data with billions of connections, we used the data structure introduced by Ganglberger et al. in 2019[27]. The high query speed is reached by sorting rows and columns of a connectivity matrix by their location in space along a space filling curve[89], so that rows and columns that represent connections that are close together in the 3D reference space are also close together on the matrix axes. This makes reading local connectivity (e.g. the connectivity of a brain region) from the hard-drive extremely efficient, since it benefits from read-ahead paging of the operating system to reach near-sequential reading speed[27].
- *Imaging data*: For computing the mean expression of a volume of interest (VOI), for example a brain region, we used spatial indexing on volumetric images similar to Schulze[59]. Here, the imaging data are not stored per image, but per voxel: For each voxel in the reference space, the data of all images at the voxel's position are stored together (i.e. on the physical hard-drive). Furthermore, we order these per-voxel data along a space filling curve[89], which allows data points in close proximity in the 3D reference to be stored in close proximity as well on the storage. The expression of the voxels of a VOI can then be read block-wise from the hard-drive, which is more efficient than reading each image individually due to read-ahead paging of the operating system[59].
- *Sample-based data*: For sample-based data, such as RNA sequencing and microarray gene expression data, we used a similar approach as for imaging data. Here, for each sample in our resource, we used the sample's mapping to the reference space (Section Mapping to a common reference space) to get the sample's location. Based on these locations, we ordered samples along a space-filling curve and stored them on the hard drive. This means, that if a certain VOI is queried for gene expression, all relevant samples of all datasets are stored close-together. As a consequence, they can be retrieved block-wise, benefiting from read-ahead paging of the operating system similar to the connectivity and imaging data approaches. We further optimized the queries by pre-aggregating samples with similar meta data, i.e. samples of the same dataset, cell type, age category, etc. This increases the query speed, since the amount of data that need to be aggregated on-the fly is reduced from thousands of individual samples to a tenth or even a hundredth of it (depending on the extent of the query).

### Data availability
BrainTrawler, including the BrainTACO resource, can be publicly accessed via braintrawler.vrvis.at. All means to set up a custom BrainTrawler instance (including a Docker image) is available at Zenodo (https://doi.org/10.5281/zenodo.10400999). This does not include the 3rd party gene expression or connectivity data. All code to include these data, as well as custom datasets into BrainTrawler is provided in Supplementary Data 1, see Section Code Availability for details.

### Code availability
The software to operate BrainTrawler consists of a multitude of services, including spatial indices for gene expression and connectivity, a graph database, a web interface, and an API to handle individual parts[26]. Hence, making the code open-source does not necessarily guarantee that it can be easily used by others. However, we believe that providing a publicly accessible, free to use instance of BrainTrawler/BrainTACO (braintrawler.vrvis.at), as well as all means to operate a custom BrainTrawler instance (see Section Data Availability) serves the scientific community best, as it allows researchers to leverage the data without the need for extensive technical knowledge. The code for the mapping and data generation of all gene expression or connectivity data of the BrainTACO resource is provided in Supplementary Data 1. In combination with the provided Docker image (see Section Data Availability), it is possible to reproduce the public instance running at braintrawler.vrvis.at, as well as to include one's own custom datasets. However, setting up a separate BrainTrawler instance may not be the most efficient approach for most researchers. VRVis, as a not-for-profit research center, is open to providing support in setting up a BrainTrawler instance in the scope of a research collaboration and is committed to further developing and improving the system to better serve the scientific community. We welcome initiatives for joint research and development projects and generally provide the software within these projects free of charge.

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

## Acknowledgements

VRVis is funded by BMK, BMDW, Styria, SFG, Tyrol and Vienna Business Agency in the scope of COMET - Competence Centers for Excellent Technologies (879730) which is managed by FFG. Research on spatial indexing was funded by the Austrian Science Fund (FWF): I 4836-B. Wulf Haubensak was supported by the Research Institute of Molecular Pathology (IMP), Boehringer Ingelheim, the Austrian Research Promotion Agency (FFG), and a grant from the European Community's Seventh Framework Programme (FP/2007-2013) / ERC grant agreement no. 311701. The extensions of BrainTrawler with the integration of transcriptomic data were funded by Boehringer Ingelheim, where we want to specifically thank Till Andlauer for his writing support, and Moritz von Heimendahl, Roberto Arban, Frank Gillardon, Sergio Picart-Armada, Gregiorio Alanis-Lobato, and Yasin Kaymaz for the selection and assistance with the datasets. Florian Ganglberger performed his work for the paper as PostDoc employed at VRVis, but finalized and submitted the paper as Principal Scientist at Boehringer Ingelheim RCV GmbH & Co KG. Furthermore, we want to thank Piotr Radkowski and Adam Filip (Ardigen) for preprocessing, Nicolas Swoboda (VRVis) for web-development support, as well as Sophia Ulonska and Johannes Novotny (VRVis) for proof reading and support during the review process.

## Author contributions

F.G. was the main author behind the development of BrainTrawler, including its conception and the mapping and processing of the datasets. F.G. also played a significant role in the creation of BrainTrawler LITE, a task shared with M.T. Both F.G. and M.T. were involved in the coding and development of BrainTrawler, M.T. for BrainTrawler LITE. D.K. and W.H. created the case study, contributing to its discussion and interpretation. Their insights were crucial to the understanding and analysis of the study. J.H.-L. was responsible for the anatomical mapping of the datasets to the reference space. In addition to this, J.H.-L. also provided writing support, ensuring the clarity and coherence of the paper. F.G., N.L., and F. F.-A. conceived BrainTACO as a resource. N.L and F.F.-A.'s input was also valuable in the development of BrainTrawler LITE. K.B. played a supervisory role throughout the project, providing guidance and oversight.

## Competing interests

The authors declare no competing interests.
