## [Peer Review File · Communications Biology]

Reviewers' comments:

Reviewer #1 (Remarks to the Author):

The article presents a novel tool to integrate transcriptomic and connectivity data across modalities and species by bringing them into a comparable space. I think this article brings interesting ideas and is a valuable tool for the scientific community. I have a few comments and questions that I hope the answers will help future readers of this article. My only real concern is the closedness of the software, as I think this will hinder its development and use by the community.

Main comments:

2.2 I am not familiar with this use of the term "ontology". Is it another term for a parcellation of the brain? Maybe adding a small definition of an ontology in this context would help some readers.

Why is structural connectivity only used for mice and functional connectivity for humans? Was the other modality not available for each? Is it possible to use the other modality with another dataset in brainTACO? I understand now, at the end of the article, that the connectivity data was acquired by injection, and thus it is not possible on humans. Then, I would suggest making that clear earlier in the paper. Also, would it be possible/useful to add structural connectivity data from DWI-derived imaging? I do not require that from the authors, but this is more of a question and potential future development.

2.3: "which can be either an arbitrary manually defined area" Is it possible to draw directly on the application?

2.4: "GWAS"? I had to google it. Is it a really common term that can stay an abbreviation in the text?

"Interestingly, human FC is not significantly correlated to mouse SC, suggesting relevant functional differences between species and/or connection modality (Figure 6b)." It is already not trivial to correlate SC to FC in humans; was it really expected between two vastly different species? Maybe the jump between species AND between modalities is too great of a leap to find a simple correlation.

My last comment concerns the absence of open source. Why that choice? I believe the scientific community has been pushing as a whole for more transparency and collaboration. I think closed-source software is detrimental to the community and the authors themselves as it prevents many people from using it and prevents community reviews and improvements.

Reviewer #2 (Remarks to the Author):

The authors present a new toolbox, BrainTACO, which allows the display and analysis of data from

genetics, functional connectivity, structural connectivity. The aim is to offer a single framework through which researchers can interact with such data and integrate results from the multiple modalities without requiring extensive coding skills. I find the aim of the toolbox is quite good and has potential to help researchers cross subfield boundaries by making all the data accessible in the same framework.

I found that the overall approach to integrate these multiple modalities robust, and only have a few comments to make. However, I was unable to access the toolbox online via the provided link – after entering the reviewer login info, I received a “502 Bad Gateway” error.

(1) I only have one major comment, and this relates to my expertise, which is neuroimaging: how was the HCP FC data mapped to MNI space exactly? The authors say that, for each grayordinate of the HCP dense connectivity data, they retrieved the closest voxel (within 1mm) of the MNI template. This approach does not yield accurate mapping between the two spaces unless the authors used grayordinates of the MNI template to perform the mapping. If the mapping was instead based on the typically used canonical brain surface provided with HCP data, i.e., the ‘fsaverage’ brain, then this mapping would not be accurate. If this is the case, then the solution is very simple: perform the same mapping but ensure that the dense connectivity data is represented on the anatomical surface of the MNI template. This is easy to do as the data is released in the space of the ‘32k_FS_LR’ surface, which means that it can readily be mapped onto a different anatomy (e.g., individual brain, MNI brain, etc.) without the need for spatial normalisation. The only thing needed would be the cortical surface of the MNI template in the same surface space (32k_FS_LR). Could the authors confirm that this is indeed the approach they followed? If not, and the authors would like to use the MNI brain cortical surface (resampled to the space of 32k_FS_LR) for this mapping, I am happy to provide this surface as I have already created it in the past.

Other minor comments

(2) Page 7, section 2.3, line 1: “web-based framework visual analytics framework” (repetition).

(3) The font size in most figures is very small and sometimes hard to read.

(4) Page 10: the entire section describing Fig. 4 should go into the caption of that figure. The figure itself can be difficult to understand without the text, so moving this description into the caption will help the reader instead of having to go back and forth between the figure and text.

(5) Page 10, paragraph 2: “we extended BrainTrawler’scapability” (typo: missing space).

Reviewer #3 (Remarks to the Author):

This paper presents an online data resource, BrainTACO that integrates multimodal neurobiological data onto a common reference space, enabling the exploration of relationships among genes, brain cell types,

regions, and circuits. It extends previous BrainTrawler, a web-based tool, and offers comparative visualizations to identify genetic drivers of connectivity, accelerating research and reducing manual data aggregation efforts for neuroscientists.

In general, the study presents an interesting resource to link genes and cell types to various brain regions as well as connectome. However, many details, particularly in the methods section, are insufficiently described, making it difficult to evaluate the rigor of this work. The rationales for some calculations and metrics such as using mean gene expression are also unclear. Additionally, the manuscript requires further editing to improve clarity. Here, I just highlight some major concerns.

- It is unclear why average gene expression can be used to present cell type specific expression. The study neglects state-of-the-art single-cell analysis techniques, such as the identification of differentially expressed genes.
- Also, it remains unclear if the study takes in account batch effects from gene expression datasets from different studies (esp. bulk tissue vs. single cell).
- Statistical approaches were not well justified. For instance, it is unclear whether p-values were corrected for multiple testing.
- Since Figure 3 looked at gene correlations (co-expression), co-expressed genes should be considered for cell-type or region specificity, which provides more mechanistic insights like gene regulation than individual genes. Additionally, it is worth noting that Figure 3 is not referenced in the main text, and therefore, its results are not discussed.
- Moreover, the study lacks independent validations of linked cell-type/region specific genes.
- The manuscript would benefit from harmonizing the writing style across different sections to improve overall clarity. Also, any typos (e.g., 20.000 genes) should be corrected.

Reviewer #1:

The article presents a novel tool to integrate transcriptomic and connectivity data across modalities and species by bringing them into a comparable space. I think this article brings interesting ideas and is a valuable tool for the scientific community. I have a few comments and questions that I hope the answers will help future readers of this article. My only real concern is the closedness of the software, as I think this will hinder its development and use by the community.

Main comments:

2.2 I am not familiar with this use of the term "ontology". Is it another term for a parcellation of the brain? Maybe adding a small definition of an ontology in this context would help some readers.

A brain ontology is a structured representation of knowledge about the brain, including its anatomical structures. It provides a standardized vocabulary and hierarchical organization of brain-related concepts, which we use to hierarchically annotate the underlying brain parcellation. We added the definition to the "Main" section.

Why is structural connectivity only used for mice and functional connectivity for humans? Was the other modality not available for each? Is it possible to use the other modality with another dataset in brainTACO? I understand now, at the end of the article, that the connectivity data was acquired by injection, and thus it is not possible on humans. Then, I would suggest making that clear earlier in the paper. Also, would it be possible/useful to add structural connectivity data from DWI-derived imaging? I do not require that from the authors, but this is more of a question and potential future development.

We chose the two modalities because of their availability and resolution. The structural connectivity for the mouse was already registered to the same reference space as the parcellation, since it was provided by the Allen Institute. We were looking for something similar for the Human, which (as mentioned by the reviewer) was not available. Hence, we were looking for an approximation such as resting-state connectivity, with both high resolution, and on the same reference space as the human parcellation. Therefore, we went with the Human Connectome Project. We added the information to the section. BrainTrawler is inherently generic, in principle one can add indefinitely more structural/functional connectivity matrices, as long as they are registered to the same reference space.

2.3: "which can be either an arbitrary manually defined area" Is it possible to draw directly on the application?

Yes, manually means in this case drawing via a brush tool.

2.4: "GWAS"? I had to google it. Is it a really common term that can stay an abbreviation in the text?

We replaced GWAS with the full name "genome-wide association studies"

"Interestingly, human FC is not significantly correlated to mouse SC, suggesting relevant functional differences between species and/or connection modality (Figure 6b)." It is already not trivial to correlate SC to FC in humans; was it really expected between two vastly different species? Maybe the jump between species AND between modalities is too great of a leap to find a simple correlation.

We agree, a correlation across connection modality and species is not necessarily expected. We adjusted the sentence accordingly ("Interestingly" to "As expected") in Section 2.4

My last comment concerns the absence of open source. Why that choice? I believe the scientific community has been pushing as a whole for more transparency and collaboration. I think closed-source software is detrimental to the community and the authors themselves as it prevents many people from using it and prevents community reviews and improvements.

Thank you for your valuable feedback on our paper, "BrainTACO." We appreciate your concern regarding the absence of open source for BrainTrawler. We would like to address your comment and provide our rationale for this choice.

Firstly, it is important to note that BrainTrawler is not the primary focus of this paper, as it was published before. The current paper focuses on the BrainTACO resource, which not only extends BrainTrawler's capabilities but also provides easily accessible holistic brain gene expression and connectivity data. However, we understand the importance of transparency and collaboration in the scientific community and would like to clarify our position on this matter.

While making the code open source is a step towards transparency, it does not necessarily guarantee that it can be easily used by others. BrainTrawler is developed and maintained by VRVis, a research institute with several software developers dedicated to its continuous improvement. The software consists of a complex structure of services, including spatial indices for gene expression and connectivity, a graph database, a web interface, and an API to handle individual parts. These components require expertise to deploy and maintain, which may not be readily available to all researchers.

However, we believe that providing a publicly accessible instance of BrainTrawler/BrainTACO (braintrawler.vrvis.at) serves the scientific community best, as it allows researchers to leverage the data without the need for extensive technical knowledge. Password protected access will be removed latest after the peer-review process (username: "reviewer" and password: "eeZ6ume1kePoodoo"). To further support collaboration and accessibility, we have now uploaded means to setup a BrainTrawler instance to Zenodo ([doi: 10.5281/zenodo.10400999](https://doi.org/10.5281/zenodo.10400999)), including a Docker image of the system that is publicly hosted, as it exceeds the size limit for supplementary data. However, we do not advise setting up a separate BrainTrawler instance, as it may not be the most efficient approach for most researchers.

VRVis, as a not-for-profit research center, is open to providing support in setting up a BrainTrawler instance in the scope of a research collaboration and is committed to further developing and improving the system to better serve the scientific community. We welcome initiatives for joint research and development projects and generally provide the software within these projects free of charge.

Furthermore, we want to point out that all the code regarding mapping to a common reference space is provided in the supplementary material (Supplementary Data 1). We have further extended the data to include all code to generate each individual BrainTACO dataset for BrainTrawler integration. In combination with the Docker image, it should be, in principle, possible to operate one's own instance of BrainTrawler/BrainTACO, including custom datasets.

We hope this response addresses your concerns and demonstrates our commitment to supporting the scientific community through our work on BrainTACO and BrainTrawler.

Reviewer #2:

The authors present a new toolbox, BrainTACO, which allows the display and analysis of data from genetics, functional connectivity, structural connectivity. The aim is to offer a single framework through which researchers can interact with such data and integrate results from the multiple modalities without requiring extensive coding skills. I find the aim of the toolbox is quite good and has potential to help researchers cross subfield boundaries by making all the data accessible in the same framework.

We want to thank the reviewer for the positive feedback on BrainTACO and recognizing its potential to help researchers integrate and interact with multi-modal data in a user-friendly framework.

I found that the overall approach to integrate these multiple modalities robust, and only have a few comments to make. However, I was unable to access the toolbox online via the provided link – after entering the reviewer login info, I received a “502 Bad Gateway” error.

Unfortunately, it seems that due to a problem during server maintenance, the service was temporarily unavailable. To avoid this in the future, BrainTrawler/BrainTACO is now hosted in a separate, dedicated instance, that is publicly accessible, and free to use at braintrawler.vrvis.at. Password protected access will be removed latest after the peer-review process (username: "reviewer" and password: “eeZ6ume1kePoodoo”).

(1) I only have one major comment, and this relates to my expertise, which is neuroimaging: how was the HCP FC data mapped to MNI space exactly? The authors say that, for each grayordinate of the HCP dense connectivity data, they retrieved the closest voxel (within 1mm) of the MNI template. This approach does not yield accurate mapping between the two spaces unless the authors used grayordinates of the MNI template to perform the mapping. If the mapping was instead based on the typically used canonical brain surface provided with HCP data, i.e., the ‘fsaverage’ brain, then this mapping would not be accurate. If this is the case, then the solution is very simple: perform the same mapping but ensure that the dense connectivity data is represented on the anatomical surface of the MNI template. This is easy to do as the data is released in the space of the ‘32k_FS_LR’ surface, which means that it can readily be mapped onto a different anatomy (e.g., individual brain, MNI brain, etc.) without the need for spatial normalisation. The only thing needed would be the cortical surface of the MNI template in the same surface space (32k_FS_LR). Could the authors confirm that this is indeed the approach they followed? If not, and the authors would like to use the MNI brain cortical surface (resampled to the space of 32k_FS_LR) for this mapping, I am happy to provide this surface as I have already created it in the past.

We are in line with the reviewer’s assessment, we concur that this not clear form the text. As proposed by the reviewer, we use grayordinates in MNI template space to perform the mapping: The human connectome project provides their own version of the 32k_FS_LR surface (S900.L.white_MSMA11.32k_fs_LR.surf.gii and S900.R.white_MSMA11.32k_fs_LR.surf.gii) in grayordinate space, which can be used via the Human Connectome Project’s Connectome Workbench to extract the coordinates of the surface vertices in the MNI template’s space (see `Supplementary_Data_1_Mapping_And_Data_Generation_Code.zip\code\generating_connectivity_matrices\Human_WU-Minn_HCP_RestingState\getVoxelAndVerticesToMNISpaceMapping.m`). As a result, we have indeed the grayordinates of the HCP FC data mapped to the MNI template’s coordinate space. The retrieval of the closest voxel (within 1mm) is merely a reference to the 1mm resolution of the MNI152 template we use as the reference space: The coordinates we retrieve from the Connectome Workbench are sub millimeter resolution, so we need to map it to the closest voxel on the 1mm “grid” (i.e. each voxel gets the connectivity of the closest vertex/voxel coordinate assigned, see `Supplementary_Data_1_Mapping_And_Data_Generation_Code.zip\code\generating_connectivity_matrices\Human_WU-Minn_HCP_RestingState\generate_connectivity_matrix_file.R`). We completely rephrased the description in the text.

Other minor comments

(2) Page 7, section 2.3, line 1: “web-based framework visual analytics framework” (repetition).

(3) The font size in most figures is very small and sometimes hard to read.

(4) Page 10: the entire section describing Fig. 4 should go into the caption of that figure. The figure itself can be difficult to understand without the text, so moving this description into the caption will help the reader instead of having to go back and forth between the figure and text.

(5) Page 10, paragraph 2: “we extended BrainTrawler’s capability” (typo: missing space).

We made the suggested adjustments to the text, and added detailed caption to Figure 4. Furthermore we made the following adjustments to figures to made the text better readable:

Figure 2: Increased font size as much as possible

Figure 3: Rearranged figure, removed unreadable text and increased font size

Figure 4 and 5 are screenshots, so font size was not improved. High resolution images are provided in Supplementary Data 3

Figure 6: Increased font size

Figure 7: Increased font size

Figure 8: Figure size increased

Reviewer #3:

This paper presents an online data resource, BrainTACO that integrates multimodal neurobiological data onto a common reference space, enabling the exploration of relationships among genes, brain cell types, regions, and circuits. It extends previous BrainTrawler, a web-based tool, and offers comparative visualizations to identify genetic drivers of connectivity, accelerating research and reducing manual data aggregation efforts for neuroscientists.

In general, the study presents an interesting resource to link genes and cell types to various brain regions as well as connectome. However, many details, particularly in the methods section, are insufficiently described, making it difficult to evaluate the rigor of this work. The rationales for some calculations and metrics such as using mean gene expression are also unclear. Additionally, the manuscript requires further editing to improve clarity. Here, I just highlight some major concerns.

We appreciate the reviewer's valuable suggestions and feedback on our manuscript. We have carefully proofread the manuscript and incorporated the reviewer's recommendations, which we believe have significantly improved the readability and overall quality of the paper.

Furthermore, we want to point out that that BrainTrawler/BrainTACO is publicly accessible, and free to use at braintrawler.vrvis.at. Password protected access will be removed latest after the peer-review process (username: "reviewer" and password: "eeZ6ume1kePoodoo").

- It is unclear why average gene expression can be used to present cell type specific expression. The study neglects state-of-the-art single-cell analysis techniques, such as the identification of differentially expressed genes.

Average gene expression provides an overview of the predominant gene activity within a specific cell type/brain region. It allows for a simplified representation of complex gene expression patterns, making it easier to use visual analytics approach to compare the key differences between cell types/brain regions without losing essential information.

We agree with your point about the importance of single-cell analysis techniques and would like to clarify that our intention was to adopt a visual analytics approach to make the analysis as simple and accessible as possible. For this, we have implemented a parallel coordinate system to compare the expression of different brain regions and cell types across datasets. This can be seen in Figure 4c and 5c, which serve as visual analytics approaches to differential gene expression.

Furthermore, we would like to highlight that our study does provide cell type and region-specificity. This approach essentially provides differential gene expression to all other cell types or the whole brain, allowing for a more comprehensive understanding of the gene expression landscape.

We hope that these clarifications address your concerns. We added a clarification to the Section 2.3 (third paragraph).

- Also, it remains unclear if the study takes in account batch effects from gene expression datasets from different studies (esp. bulk tissue vs. single cell).

In general, we account for batch effect in the single cell data via preprocessing with the Seurat package (see “Methods” section, subsection “Data Preprocessing and Normalization”), except for STAB datasets, for which this had been already performed consistently. For bulk we directly took the data from GTEX and BrainSpan, which according to the data sets’ publications, were already preprocessed and accounted for batch effects. We want to point out, that in our study, we focus on the ranking of gene expression (i.e. how a gene’s expression is ranked relative to all other genes) rather than absolute gene expression values. This acts as a form of normalization to adjust for batch effects. We added a clarification to the discussion section.

- Statistical approaches were not well justified. For instance, it is unclear whether p-values were corrected for multiple testing.

We intended to use the p-value as a discovery tool to threshold correlation coefficients, not to state significance of correlation coefficients of individual genes per se. However, we agree that this might be misleading and therefore adapted our analysis to cover the highest and lowest 1% of correlation coefficients, respectively. In doing so, we conclude that the results obtained are largely similar, again suggesting that patterns observed in the mouse transfer to some degree to Human (Figure 6c,d,f)

- Since Figure 3 looked at gene correlations (co-expression), co-expressed genes should be considered for cell-type or region specificity, which provides more mechanistic insights like gene regulation than individual genes. Additionally, it is worth noting that Figure 3 is not referenced in the main text, and therefore, its results are not discussed.

We agree that mechanistic insights like gene regulation can provide more valuable information than individual genes. However, we would like to point out that the different datasets used in our study cover various different cell types and regions. As a result, the cell-type and region specificity of co-expressed genes would not be directly comparable across these datasets. Hence, the approach via gene correlation. We want to point out that Figure 3 is referenced Section 2.1 , for which we added a more detailed discussion.

- Moreover, the study lacks independent validations of linked cell-type/region specific genes.

We presume that this point pertains to genes with highest/lowest correlations with connectivity. We would like to point out that these genes were put into context of existing data using the OpenTargets database (Figure 6d, Supplementary Table 5). We believe that further experimental validations are beyond the scope of this study and atypical in this setting.

- The manuscript would benefit from harmonizing the writing style across different sections to improve overall clarity. Also, any typos (e.g., 20.000 genes) should be corrected.

We had the manuscript proofread once more by an external party to ensure a higher quality and overall clarity.

REVIEWERS' COMMENTS:

Reviewer #2 (Remarks to the Author):

I thank the authors for the clarifications and the changes they made to the manuscript. From my perspective, I am happy with the current version.

Reviewer #4 (Remarks to the Author):

Concerns are properly addressed except that I have one more question.

The authors claimed that there were no structural connectivity for humans. Thus they used the functional connectivity data from Human Connectome Project (HCP). However, structural connectivity data were also collected in HCP and provided to the scientific community. Any relevant discussion would be helpful.

Reviewer #2:

I thank the authors for the clarifications and the changes they made to the manuscript. From my perspective, I am happy with the current version.

Thank you for your valuable feedback and time; we are delighted to hear that you are satisfied with the current version of our manuscript

Reviewer #4:

Concerns are properly addressed except that I have one more question.

The authors claimed that there were no structural connectivity for humans. Thus they used the functional connectivity data from Human Connectome Project (HCP). However, structural connectivity data were also collected in HCP and provided to the scientific community. Any relevant discussion would be helpful.

We appreciate the reviewer's valuable suggestions and feedback on our manuscript. We acknowledge that the Human Connectome Project (HCP) does provide structural connectivity data. However, at the time of our research, the structural connectivity data was not as readily preprocessed as the resting state fMRI (all subjects combined into one grayordinate-level connectivity matrix on a common reference space), which is one reason why we opted for the latter. Since the submission of our paper, we have noticed that a region-level matrix for structural connectivity has been made available by Škoch, A., Reháček Bučková, B., Mareš, J. et al. Human brain structural connectivity matrices—ready for modelling. *Sci Data* 9, 486 (2022). However, this is a 90x90 region-level matrix, while the resting-state fMRI provides voxel-level (grey-ordinate-level) data. As our tool, BrainTrawler, was designed to support even voxel-level connectivity, we chose the resting-state fMRI connectivity for showcasing higher resolution workflows that go beyond region-level.

We now included this relevant discussion in our manuscript ("Section Methods - Data Preprocessing and Normalization – Resting-State Functional Connectivity Data", and a mention in Section Results - "We selected both the mouse structural connectivity data and the human resting-state functional connectivity data for their high resolution and compatibility with the common reference spaces (see Section Methods - Data Preprocessing and Normalization for more details).").